# Evidence for highly variable land use but a stable climate in the southwest Maya lowlands

Benjamin Gwinneth<sup>1,2</sup>, Kevin Johnston<sup>3</sup>, Andy Breckenridge<sup>4</sup>, and Peter MJ Douglas<sup>2</sup>

<sup>1</sup>Département de géographie, Université de Montréal, Montréal, H2V 0B3, Canada

<sup>2</sup>Department of Earth and Planetary Sciences, McGill University, Montréal, H3A 0E8, Canada

<sup>3</sup>Independent Scholar, Columbus, 43214, United States of America

<sup>4</sup>Natural Sciences Department, University of Wisconsin-Superior, Superior, 54880, United States of America

Correspondence to: Benjamin Gwinneth (benjamin.gwinneth@umontreal.ca)

Abstract. The lowland Maya of Mesoamerica were affected by multiple environmental stresses throughout their history, and many experienced a major demographic and political decline, or collapse, during a period of inferred intense multidecadal drought, approximately 1200- and 1000-years BP. Given regional variation in the timing and character of the collapse (Demarest et al. 2004; Hodell et al. 2007; Webster et al. 2007; Kennett and Beach 2014; Douglas et al. 2015), much remains to be discovered about the complex interactions between climate and society in the Maya lowlands. To this end, we combine carbon and hydrogen isotopic analyses of leaf wax *n*-alkanes with quantification of faecal stanols and polycyclic aromatic hydrocarbons from a lake sediment core from the southwest lowlands to assess whether 1) palaeoecological evidence of land use is related to population change; and 2) whether population and land use are linked to changing precipitation. Our data reveal a transition from generally more intense fire use and C<sub>4</sub> plant agriculture during the Preclassic (3500-2000 BP) to dense populations and reduced fire use during the Classic (1600-1000 BP). This is consistent with other evidence for a more urbanised and specialised society in the Classic. We do not find evidence of drought in the hydrogen isotope leaf wax record (δD<sub>lw</sub>), implying that local drought was not a primary driver of observed variability in land use or population change in the Classic-period southwestern lowlands.

#### 1 Introduction

The lowland Maya of Mesoamerica experienced a major demographic and political decline, sometimes referred to as a collapse, during a period of intense multidecadal drought, between approximately 750 and 900 CE (Hodell et al. 1995; Rosenmeier et al. 2002; Aimers 2007; Medina-Elizalde et al. 2016). Archaeological and paleoecological research suggests variation in the causes and timing of sociocultural declines (Brenner et al. 2002; Wahl et al. 2014; Carleton et al. 2014; Beach et al. 2015; Akers et al 2016; Medina-Elizalde et al. 2012). Itzan, an ancient population centre in the southwest Maya lowlands (Johnston 2001; Johnston et al. 2006), is in the relatively small catchment of Laguna Itzan, where a sediment core has been previously

described (Breckenridge 2000; Douglas et al. 2018; Keenan et al. 2021). We applied different geochemical proxies to the same core to reconstruct how precipitation, fire and agriculture varied over time, and we compare these with previously published stanol data on human waste and demographic change.

Figure 1. a) 3D relief and rainfall map of Central America with location of Itzan and Lakes Chichancanab, Salpetén, and Kail. Colours are an average of annual summed precipitation in mm (CHIRPS: Climate Hazards Center InfraRed Precipitation with Station data) between 1989 and 2019. b) Regional map of the Maya area is inset bottom right.

Polycyclic aromatic hydrocarbons (PAHs) are produced from the incomplete combustion of organic matter in conditions of oxygen depletion and can be preserved in lake sediments, where their concentration can be used as a proxy for past fire frequency and intensity (McGrath et al. 2003; Conedera et al. 2009; Denis et al. 2017). In this study PAH analyses are applied to understand the timing and extent of fire used for local agricultural land clearance or building. Fire was used as a method of deforestation, and in the use of swidden, or slash-and-burn, agriculture (Brittingham et al. 2019). Leaf wax lipids are synthesised by plants as chemical barriers, to inhibit water loss, and to provide protection from disease and UV light

(Diefendorf & Freimuth, 2017). Leaf waxes record the stable carbon isotope ratio ( $\delta^{13}$ C) of vegetation, which is strongly influenced by the photosynthetic pathway of the plant, namely differentiating C<sub>3</sub> plants (most trees, shrubs, and forbs) and C<sub>4</sub> plants, which include many tropical grasses, such as maize, which the Maya cultivated (Wright & White, 1996; Feakins & Sessions, 2010).  $\delta^{13}$ C of plant waxes represents the ratio of C<sub>3</sub> to C<sub>4</sub> terrestrial plants and can therefore indicate when there was a greater amount of maize cultivation in the catchment. Plants waxes also record the isotopic composition of hydrogen ( $\delta$ D) in plant tissue (Sachse et al. 2012; Sachse et al. 2015). Precipitation  $\delta$ D in the low-elevation tropics is largely determined by the amount of rain, such that with increased rainfall, the  $\delta$ D is lower (Sachse et al. 2015; Niedermeyer et al. 2016; Douglas et al. 2015). This water is used in the biosynthesis of leaf waxes, thereby preserving isotope composition, although it is subject to a large biosynthetic fractionation factor that can vary between plant groups (Sachse et al. 2015). In the lowlands, spatial variation in the  $\delta$ D<sub>lw</sub> of leaf waxes appears to be strongly controlled by soil water evaporation, and, therefore, past variation is likely a combined effect of precipitation isotope variability and soil water evaporation (Douglas et al. 2015).

We combine these new data with previously published palaeoclimate records for Lakes Chichancanab and Salpetén (Fig. 1; Douglas et al. 2015), and faecal stanols as a proxy for human population (Keenan et al. 2021). PAH concentrations may have increased with increasing leaf wax  $\delta^{13}C_{lw}$ , as land was cleared for maize agriculture (McGrath et al. 2003; Sluyter & Dominguez, 2006). We also hypothesised that agricultural expansion, indicated by increasing  $\delta^{13}C_{lw}$  values, and population growth, indicated by increased concentrations of faecal stanols, occur in response to amenable climate conditions, indicated by decreasing  $\delta D_{lw}$  values. Further, we hypothesised that Terminal Classic population decline was associated with more positive  $\delta D_{lw}$  values, which indicate dry climate conditions indicative of drought, as seen in other areas of the Maya lowlands.

# 2 Results

80

We identify five events distinguished by changes in the four proxy records (Fig. 2). Event 1, at 4000 years BP, prior to archaeological or molecular evidence of human presence, is marked by a pulse of enriched  $\delta^{13}C_{lw}$ . Event 2, characterised by evidence of fire, potentially for land clearance, at 3200 years BP, coincides with a smaller 1 ‰ shift to more positive  $\delta^{13}C_{lw}$  values and an increase in coprostanol, a marker for human population. There is no consistent relationship between  $\delta D_{lw}$  and  $\delta^{13}C_{lw}$  in the record, but the second major peak in  $\delta^{13}C_{lw}$  (-30 ‰) at 2900 years BP (Event 3) is coeval with the wettest period in the record, and it precedes a ~900-year period of fluctuating  $\delta^{13}C_{lw}$  values and inputs of coprostanol, The interval of highly variable land use and coprostanol concentrations culminates at 1900 years BP with the most positive  $\delta^{13}C_{lw}$  values (-25 ‰) and Total PAH concentrations of 3.1 µg g<sup>-1</sup> OC, marking Event 4, and it is followed by a ~500-year period of low coprostanol and PAH concentrations. Coprostanol input then increases gradually peaking at 1100 years BP (Event 5) co-eval with a  $\delta^{13}C_{lw}$  peak of -25‰ and 1.7 µg g<sup>-1</sup> OC Total PAH. The Terminal Classic, between 1140-1000 years BP, is marked by an abrupt

decline in coprostanol and PAHs and more negative  $\delta^{13}C_{lw}$ . The variability of the  $\delta D_{lw}$  record defined by maximum versus minimum is only 22 ‰ at Itzan, whereas it is 44 ‰ at Chichancanab and 55 ‰ at Salpetén to the northeast (Fig. 3).

Figure 2. a) Coprostanol concentrations normalised to organic carbon ( $\mu$ g/g of OC), with raw data in grey and the purple line represents a 100-year moving average; b) Total PAH in grey with the red line a moving average ( $\mu$ g/g of OC);  $\delta D_{lw}$  of  $C_{29}$  n-

alkanes in grey and yellow a moving average (‰); and d)  $\delta^{13}C_{lw}$  of  $C_{29}$  n-alkanes in grey with moving average in green (‰) with %  $C_4$  plants from a mixing model. For  $\delta D_{lw}$  and  $\delta^{13}C_{lw}$  (c and d) the grey envelope is the pooled standard error. Note that not all samples analysed for  $\delta^{13}C_{lw}$  could be analysed for  $\delta D_{lw}$ . 5 "events" have been identified based on peaks that deviate from the baseline record as well as peaks that co-occur across different proxies and are shown as pink bars. Maya time periods are shown below. TC=Terminal Classic; LC=Late Classic; EC=Early Classic; EC=Early Classic; EC=Early Preclassic; EC=Early Preclassic.

#### 3 Discussion

#### 3.1 Progressive changes in modes of land use over time

Enriched δ13Clw in Event 1 (roughly 4000 BP) could relate to major changes in catchment vegetation during a particularly short period of occupation. That this peak was short-lived suggests that the population was likely itinerant, possibly mobile hunter gatherers. Zea mays L., maize pollen, is found in southeast Mexico near the mouth of the Usumacinta River by 5000 years BP (Sluyter & Dominguez, 2006), and carbon isotopes of human skeletons suggest that maize was a staple in Belize between 4700- and 4000-years BP (Kennett et al. 2020). This evidence suggests the presence of an Archaic (pre-agricultural) population in the southwest lowlands.

Event 2, characterized by evidence of fire, potentially for land clearance, at 3200 years BP, coincides with a small 1 ‰ shift to more positive δ13Clw values and an increase in coprostanol concentrations, a marker for human population (Keenan et al. 2021; Vachula et al. 2019; D'Anjou et al. 2021; Keenan et al. 2022). Thus, this period, the late Early Preclassic, was a time of significant landscape change in the catchment of Itzan. The change in δ13Clw shows burning for land clearance—perhaps for C3 plants or to attract deer to forest edges. Early burning, in other words, was not always associated with extensive C4 plant domestication or agriculture. The data provide the first evidence of an Early Preclassic population in the southwest lowlands. Event 3, dating to the early Middle Preclassic period, indicates population variation and periodic burning. Overall, this pattern is suggestive of populations moving in and out of the area, with variable degrees of associated burning and land clearance, and possibly different farming techniques and subsistence strategies with variable dependence on maize agriculture. Coevally at nearby Ceibal and at the more distant Aguada Fenix (Inomata et al. 2015; Inomata et al. 2020) non-sedentary populations for the first time aggregated to build substantial public works. Perhaps the paleoecological evidence reveals a similar process at Itzan. Coeval ceramics provide the earliest archaeological evidence of a human presence at Itzan (Johnston 2006).

The most positive  $\delta$ 13Clw values (-25 ‰) occur during the Late Preclassic Event 4 suggesting significant agricultural activity, and the Total PAH concentrations of 3.1  $\mu$ g g-1 OC suggest significant burning. Again, these peaks are short-lived and are followed by a ~500-year period of low coprostanol and PAH concentrations, indicative of depopulation and minimal land use.

This hiatus in coprostanol and PAH input coincides with a Terminal Preclassic through Early Classic decline at nearby Ceibal

(Willey et al 1975; Inomata et al. 2017) and Altar de Sacrificios (Smith 1972). At Itzan and the neighbouring Chak Akal (Johnston 2006), this is evident archaeologically in an almost complete absence of Early Classic ceramics.

Following this period, population then increased gradually, peaking at 1100 years BP (Event 5, Late Classic), which is also the local and regional population maximum inferred by archaeological evidence (Johnston 2004). Notably,  $C_4$  plant coverage declined during the early part of the Classic (1750-1300 BP), when human population was increasing, indicating there is not a linear relationship between  $C_4$  plant agriculture and catchment population density. This may imply that agriculture was intensified in the catchment around the time that population peaked. Lower  $\delta^{13}C_{lw}$  values than the preceding period, except for the peak at 1100 years BP, suggest that Classic population growth was accompanied by a change in agricultural production. A coeval reduction in coprostanol, despite the population maximum inferred by archaeological evidence, was discussed in Keenan et al. (2021) and explained as the result of a transition in agricultural strategy, including efforts to reduce soil erosion. Similar strategies have been observed elsewhere, for example at Tikal, such as ridge and furrow cultivation, the use of vegetation along the margins of water sources, intensive dooryard gardening and orchards, and extensive farming in outfield areas (Fedick et al. 2024).

Except for the PAH peak at 900 years BP, PAH levels never reached pre-1900 years BP levels again. This could imply that most forest had already been cleared prior to the Classic and/or that fire was not a method for land clearance during the Classic, despite larger populations and a  $\delta^{13}$ C<sub>lw</sub> peak of -25% indicative of significant, albeit short-lived, C<sub>4</sub> plant agriculture. The likely explanation is that fire was no longer a key component of agricultural production during the Late Classic and fits with the narrative of agricultural intensification described above. Later input of PAHs may, therefore, have been more protracted because of low intensity burning. The Terminal Classic, between 1140-1000 years BP, is marked by an abrupt decline in coprostanol and PAHs and more negative  $\delta^{13}C_{lw}$ , suggesting site abandonment and cessation of fire use and agriculture. This period was a time of major societal restructuring across the lowlands, hypothesised to be in response to a regional drought, although there is no significant shift in the  $\delta D_{lw}$  record (Hodell et al. 1995; Rosenmeier et al. 2002; Keenan et al. 2021). This response was the same across different sites. C4 plant agriculture and fire appear to be most prevalent during the Preclassic and may be a signal of shifting cultivation. High variability may imply different groups with different land use strategies, or variable land use by the same group. Conversely, for much of the Classic period fire and C4 plants are lower despite sustained population. This might reflect the employment of more efficient agricultural strategies following the recognition of soil degradation and/or the necessity to sustain growing populations. This variability in land use does not appear to be related to climate variability in any clear or consistent way, given the relatively muted climate variability recorded at Itzan. In this sense, Itzan contrasts with other sites where climate has been seen to drive societal and land use change (Rosenmeier et al. 2002; Hodell et al. 2001; Hodell et al. 2008).

Figure 3. δD<sub>Iw</sub> (‰) of leaf wax n-alkane records as proxies for palaeohydrological change moving southwards from a) Lake

Chichancanab; to b) Lake Salpetén; to c) Laguna Itzan; to d) Lake Kail. Maya time periods are shown below. TC=Terminal

Classic; LC=Late Classic; EC=Early Classic; TP=Terminal Preclassic; LP=Late Preclassic; MP=Middle Preclassic; EP=Early Preclassic. For Itzan (c) raw data is in blue and the vellow represents a 100-year moving average. The grey envelope is the pooled standard error.

#### 3.2 Relatively stable climate in the southwest lowlands 165

Despite a major restructuring of society, there is no evidence for drought in the  $\delta D_{lw}$  record at Itzan, though the climate signals recorded by plant waxes are expected to be broadly regional in scale. Other palaeoclimate records suggest that climate change was not uniform across Central America or within the Maya lowlands (Stansell et al. 2020; Winter et al. 2020; Duarte et al. 2021). Precipitation variability in Guatemala is thought to be controlled by changes in moisture availability associated with Atlantic and Pacific basin ocean-atmosphere dynamics and sea surface temperatures. Differences in precipitation records may 170 be explained partially by latitude; however, Itzan is only 95 km southwest of Salpetén. We suggest that differences in climate variability on small spatial scales could also be explained by local climate related to distance to the Cordillera. A band of high precipitation overlying Itzan is controlled by the Caribbean low-level jet and orographic precipitation as it is lifted over the Cordillera (Cook and Vizy, 2010). A  $\delta^{18}$ O record from Lake Kail (Fig. 3d) in the highland region similarly shows no signs of a Terminal Classic drought and is relatively muted in terms of climate variability (Stansell et al. 2020). This contrasts with 175 precipitation to the northeast, which may have been controlled more strongly by convection and, therefore, relative differences in land and ocean temperatures. In addition, it has been proposed that Maya deforestation could have reduced convective precipitation (Cook et al. 2012), and it is possible that this effect was less important in areas more strongly influenced by orographic precipitation. Finally, small changes in the zone of seasonal subsidence that causes relatively dry conditions in the 180 northern Yucatan Peninsula might have affected Lake Salpetén but would have been less likely to influence Laguna Itzan (Fig. 3). Significant differences in modern rainfall today support the idea of spatial heterogeneity in the past, and proximity to the Cordillera may drive these patterns (Fig. 1). Orographic "gaps" result in localised wind jets and affect precipitation, resulting in less climate variability in the southwest lowlands (Baldwin et al. 2021). Our data suggest that the southwest Maya lowlands had less extreme drying or climate variations than other regions, which is intriguing as this region is thought to have been where collapse began (Douglas et al. 2016).

Despite the lack of evidence for local drought, population did decline at Itzan, perhaps pointing to a regional interdependency related to drought elsewhere (Keenan et al. 2021). Societal upheaval in the central lowlands and in the Petexbatun polity, including inter-polity warfare, the collapse of royal dynasties and regional political hierarchies, population dislocations, and the disruption of regional economies and inter-regional exchange (Demarest 2004), almost certainly would have had significant pan-lowland repercussions. Complex and spatially variable patterns of climate change across the Maya lowlands would have exasperated the effects, resulting in a complex mosaic of social, political, and environmental outcomes. For example, the Rio

de la Pasion catchment, within which Itzan lies, was not completely de-populated following the Terminal Classic population decline (Johnston et al. 2001). Small inputs of coprostanol and PAHs suggest the presence of smaller numbers of people, although the largest PAH peak in the record, at 840 years BP, may be the result of wind-blown deposition from forest fires in dry landscapes to the north or locally (Vachula et al. 2022). Increasingly negative  $\delta^{13}$ Cl<sub>w</sub> values until -36.5 % suggest forest recovery to greater extents than at any point in the record, in the absence of large populations and deforestation.

## **4 Conclusions**

This study reveals the changing land use over 3300 years reflecting the process of urbanisation – a decrease in fire use and movement of maize agriculture outside of the lake catchment. Secondly, our data suggest that the southwest Maya lowlands had less extreme drying or climate variations than other regions. This is important because some archaeologists propose this is where collapse began (Douglas et al. 2016) and if drought did not occur strongly, then it implies that in the southwest lowlands, it was not a precipitating cause of depopulation or collapse. A lack of temporal and spatial high-resolution palaeorecords in the region limit our ability to fully test the hypothesis of drought-related collapse in the southwest Maya lowlands, and similar multi-proxy work in the future could help us better understand the evolution of climate and society through time.

## Methods

Sediment cores from Laguna Itzan in the southwest Maya lowlands were collected in 1997 and described by Breckenridge (2000). Two overlapping cores totalling 5.7 m of sediment were collected in 10.1 m water depth near the western shore of the lake, close to the deepest point of the lake (16.598° N, 90.4784° W). The core has been stored refrigerated at the LacCore facility at the University of Minnesota Twin Cities. The core was sub-sampled to obtain a temporal resolution of approximately 50–100-year intervals, without reference to sediment lithology. These samples were subsequently freeze-dried to remove water. Dried sediment samples were ground, weighed, added to a PTFE tube and extracted using a CEM MARS 6 microwave extractor with 10 mL of 9:1 dichloromethane:methanol. This ratio of solvents was selected after testing various methods for their extraction efficiencies using lake sediment samples (Kornilova & Rosell-Melé, 2003; Battistel et al. 2015). The MARS 6 oven was heated to 80 °C and held at that temperature for 20 minutes. The contents of the PTFE tube were then transferred to a centrifuge vial, centrifuged and the Total Lipid Extract (TLE) was transferred to an evaporating vial. 3 mL of 9:1 dichloromethane:methanol was added twice more to the centrifuge tube to ensure complete removal of extracted material. The TLE was evaporated and split into 2 fractions (a non-polar fraction and a polar fraction) using silica gel chromatography. The pipette columns consisted of 5 cm of silica gel, and 1 cm of sodium sulphate. 15 mL of hexane was eluted to collect the non-polar hydrocarbon fraction and 15 mL of methanol was eluted to collect the remaining neutral and polar fractions. The hydrocarbon fraction was analysed using gas chromatography with a flame ionisation detector (GC-FID) with a TRACE TR-

5 GC Column (60 m × 0.25 mm) at McGill University in sequence with known standards for fluoranthene, pyrene, benzo[blfluoranthene, benzo[elpyrene, and benzo[ghi]perylene (Sigma-Aldrich). The polar fraction was saponified and the same sample aliquots were used to produce the previously published stanol record. The *n*-alkane hydrogen and carbon isotope measurements were made using a Thermo Delta V Plus Isotope Ratio Mass Spectrometer (IRMS) coupled to a Thermo GC-IsoLink at McGill University, which uses a Thermo TG5Ms column (60 m × 0.25 mm × 0.25 mm). Compound-specific isotope values were calibrated to the V-SMOW and V-PDB scales using a set of n-alkane standards (Mix A6) that were measured offline in references to international standards at Indiana University (arndt.schimmelmann.us/compounds.ht). The Mix A6 standards were used to calibrate both hydrogen and carbon isotope measurements for n-alkanes and were analysed at least twice daily. Laboratory standards were also analysed to monitor for instrumental drift and assess measurement reproducibility. The laboratory standards were derived from extracts of maple (Acer sp.) leaves collected on the campus of McGill, with synthetic alkanes (Sigma-Aldrich) added to enhance the C22 and C30 n-alkane concentrations. Laboratory standards were run after every 3 samples to check for instrumental drift. Only the C<sub>29</sub> n-alkane was consistently resolved throughout the core, and therefore we focus on this measurement as it represents the signal of terrestrial plants. C<sub>29</sub> production is linked to terrestrial trees (Bliedtner et al. 2018; Aichner et al. 2018). Measurement uncertainty was estimated through replicate or triplicate analysis of each sample. Isotope data are presented as the mean of double or triple replicates with a pooled standard error shown in grey envelope (Polissar & D'Andrea, 2014; Fig. 2). The analytical standard error ranged from 3.1 to 4.4‰ for δD<sub>lw</sub> measurements and is 0.25% for  $\delta^{13}C_{lw}$  measurements.

The age model for the PAHs and faecal stanols, is based on an age model created using <sup>14</sup>C ages of terrestrial macrofossils (Douglas et al. 2018; Keenan et al. 2021) from the cores calibrated using IntCal13 (Douglas et al. 2018;Reimer et al. 2013) using a 4<sup>th</sup>-order polynomial fit with the Classical Age-Depth Modeling (CLAM) software in R (Blaauw 2010). The average age uncertainty (95% confidence interval) of the chronology is 100±30 years. The age model for the leaf wax *n*-alkanes is based on compound specific radiocarbon dating of *n*-alkanoic acids from the Itzan core. The use of the age chronology based on *n*-alkanoic acids from Itzan for the *n*-alkanes assumes that the *n*-alkanoic acids were produced at the same time as the *n*-alkanes. Because of the potential residence time of leaf waxes in sediments for up to thousands of years, this chronology is more accurate for understanding isotopic variation in leaf waxes than the age model determined from terrestrial macrofossils found in sediments (Douglas et al. 2016; Douglas et al. 2014). As we are interested in the climate at the point of formation and as it has been shown that waxes can take up to thousands of years to be deposited in lake sediments, it is necessary to use the compound specific radiocarbon age model for the waxes. For the stanols and PAHs it might also be the case that there is some long-term soil storage, but in the absence of compound specific ages we assume low residence times before deposition, and include 100-year moving averages. Faecal stanol concentrations agree broadly with population change inferred from archaeological evidence (Johnston 2006). Furthermore, for concentration measurements of PAHs and stanols, our focus is the

flux to the lake at a given point in time. Therefore, the relevant time for the paleoclimate signal is the time of deposition (terrestrial macrofossil age model), whereas for the *n*-alkane isotope data it is the time of molecule formation (compound specific age model). However, we acknowledge that long-term soil storage of stanols or PAHs would complicate their records, and that some of the PAH and δ<sup>13</sup>C<sub>lw</sub> peaks could potentially overlap, and this question deserves future attention. The relationship between stanols and sediment delivery was addressed in Keenan et al. (2021): there is no relationship between the biomarker data and the sedimentology, including magnetic susceptibility.

# Data availability

Data in this manuscript are available here https://doi.org/10.6084/m9.figshare.29497826.v1

## 265 Author contribution

BG: Conceptualization, Methodology, Investigation, Validation, Visualization, Writing - Original Draft; KJ: Resources, Writing - Review & Editing; AB: Resources, Writing - Review & Editing; PD: Conceptualization, Supervision, Funding acquisition, Writing - Review & Editing.

# 270 Declaration of competing interest

The authors declare that they have no known competing financial interests or personal relationships that could have appeared to influence the work reported in this paper.

#### Acknowledgements

We thank Bjorn Sundby for his mentorship, Tristan Grupp for GIS contributions, Thi Hao Bui for lab assistance, and the staff at LacCore for assistance with sampling archived sediment cores. We thank Nicholas Dunning and two anonymous reviewers for their comments. Funding for this project came from the Eric Mountjoy Fellowship, McGill startup funds and an NSERC Discovery Grant 2017-03902 to PMJD.

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
