# Peer review of "Evidence for highly variable land use but a stable climate in the southwest Maya lowlands"

_EGUsphere, 2025_

## Author Response (AR1)

**REVIEWER 1 (Nick Dunning)**

First, my thanks to Dr. Dunning for his thoughtful, thorough, and (thank you) enthusiastic review. I will address three concerns raised in his review.

The first pertains to my reference to pan-lowland trends during the Early Classic. Dr. Dunning remarks:

"Pan-lowland Early Classic demographic collapse noted by the authors is not a uniform phenomenon across the lowlands, nor is there a consensus among Mayanists as to its extent and whether "collapse" is an appropriate term. Citations needed to at least back up this statement. Yes, it is certainly evident in places – and probably linkable to 2nd century CE droughts, at least in places."

My original intention was to situate trends at Itzan with those found elsewhere in the Pasion drainage: i.e., what happened at Itzan resembles what happened at Ceibal and Altar de Sacrificios (and at Chak Akal, for that matter). To restore that focus I have rewritten the associated prose as follows:

"This hiatus in coprostanol and PAH input coincides with a Terminal Preclassic through Early Classic decline at nearby Seibal (Willey et al 1975; Inomata et al. 2017) and Altar de Sacrificios (Smith 1972). At Itzan and the neighboring Chak Akal (Johnston 2006), this is evident archaeologically in an almost complete absence of Early Classic ceramics. "Lines 149-152.

**Second, Dr. Dunning also remarks:**

"The lack of evidence for severe drought in the SW Maya Lowlands is intriguing and supports models of alternative causes underlying the Classic collapse." Arthur Demarest has notably championed alternative causes (e.g., escalating conflict over riverine trade routes) after noting that by some measures the collapse seems to start in the comparatively wet SW Lowlands and not in drier, presumably more drought-vulnerable areas. It would be useful to cite the 2004 edited volume (*The Terminal Classic in the Maya Lowlands: Collapse, Transition, and Transformation.* A. Demarest, P. Rice, and D. Rice, eds.) the sub-title of which speaks volumes about the ways in which the "collapse" was manifest across the lowlands – and that it was far from a simultaneous event."

In response, I have rephrased one sentence and part of a paragraph that, I hope, explicitly acknowledge the important observations made by Demarest and others regarding the character and timing of the collapse in the southwestern lowlands:

"Given regional variation in the timing and character of the collapse (Demarest et al. 2004; Hodell et al. 2007; Webster et al. 2007; Kennett and Beach 2014; Douglas et al. 2015), much remains to be discovered about the complex interactions between climate and society in the Maya lowlands." Lines 8-11.

"Despite the lack of evidence for local drought, population does decline at Itzan, perhaps pointing to a regional interdependency related to drought elsewhere (Keenan et al. 2021). Societal upheaval in the central lowlands and in the Petexbatun polity, including inter-polity warfare, the collapse of royal dynasties and regional political hierarchies, population dislocations, and the disruption of regional economies and inter-regional exchange (Demarest 2004), almost certainly would have had significant pan-lowland repercussions. Complex and spatially variable patterns of climate change across the Maya lowlands would have exasperated the effects, resulting in a complex mosaic of social, political, and environmental outcomes." Lines 235-241.

Third, Dr. Dunning refers to evidence of a persistent Postclassic population in the Itzan basin.

"The paleoecological evidence of a persistent population in the Itzan basin in the Postclassic mirrors similar findings elsewhere (e.g., around Laguna de Terminos – a large suburban reservoir at Tikal – as well as the Mucal reservoir at Yaxnohcah, and, as we are now finding at Calakmul around several reservoirs). It would be good to cite a source that documents late persistence of population in the Rio de la Pasion region."

Evidence of a Postclassic Itzan occupation is examined in our 2021 publication ("Molecular evidence for human population change associated with climate events in the Maya lowlands" but not directly in this manuscript. I concur that our 2021 article would have been strengthened by mentioning the presence of Postclassic groups at Tikal, Calakmul, and Yaxnocah—the failure to include these references is an unfortunate oversight. That said, I do refer in our 2025 manuscript to the presence of Postclassic peoples within the Pasion drainage:

"For example, the Rio de la Pasion catchment, within which Itzan lies, was not completely de-populated following the Terminal Classic population decline (Johnston et al. 2001)."

Here the reference is to paleoecological evidence of an Early Postclassic population at Laguna Las Pozas, located near Aguateca. Finally, the team plans to prepare next a manuscript that explicitly explores and summarizes the archaeological ramifications of the Itzan paleoecological data. We will be sure to include the references suggested by Dr. Dunning in that paper.

On the topic of carbon isotopes, Dr. Dunning writes:

"The authors interpret changes in the C isotope ratios into the Classic period as indicative of increasing spatial concentration and intensification of cultivation closer to Itzan itself and away from the lake margins – though also allowing for a move away from C4 plants and inclusion of more C3 plants in the agricultural mix. Both are certainly possible, though I think that increasing attention needs to be given to the importance of C3 plants in ancient Maya agriculture. Many important root crops, which are largely invisible in lacustrine pollen studies are increasingly showing up in aguadas/reservoirs (where local pollen rain predominates), and other proxies (e.g., starch grains and eDNA). Of course, the most telling example of root, arboreal and other crops is the remarkable gardens and fields of Joya de Ceren. While Ceren is not within the Maya heartland, its agricultural practices should be a red flag warning about making too many assumptions about the role of maize – and the interpretation of maize cultivation proxies. Yes, maize was almost certainly the single most important crop but it was but one of a cornucopia of cultigens.."

Thank you for this important point. The carbon isotope data tell us about the C3/C4 mix of terrestrial plants. Though it cannot tell us specifically about all cultivation, it does provide a valuable reference for when there was a great amount of maize cultivation in the catchment. This does seem to coincide with changes and burning in some instances. On lines 64-65 we have added a note to make this clearer.

On the topic of source of waxes analysed for C isotopes, Dr. Dunning writes:

"As an aside, another potential confuser for the interpretation of C isotopes might be found in the concentration of blue-green algae in reservoirs, ponds, and shallow lakes with fluctuating water levels and eutrification."

We note that the C isotope data come from C29 alkanes (derived from terrestrial plants) and as such are not input by algae. This has been made clear in the manuscript on line 235 this has been added in the methods section.

**REVIEWER 2 (Anonymous)**

First, I thank reviewer 2 for bringing to our attention the awkward statement in the Abstract regarding the interactions in the Maya lowlands between climate and society.

"1. The statement "The complex interactions between climate and society in the Maya lowlands are generally not well understood' in the Abstract and Introduction lack reference to support. The environmental models of the Terminal Classic Maya political/kingship and human-environment relationship have been widely discussed since the 1980s. There is considerable work focusing on reconstructing paleohydroclimate and climate conditions, using stalagmite and sediment core data, magnetic susceptibility, and pollen analyses, etc."

I concur with the reviewer's observation and have rephrased the sentences in question to clarify the point we had intended to make. The five references provided capture the breadth of positions taken by archaeologists and paleoecologists on the issue of Classic Maya collapse and droughts. Our rephrased statement is as follows:

"The lowland Maya of Mesoamerica were affected by multiple environmental stresses throughout their history, and many experienced a major demographic and political decline, or collapse, during a period of inferred intense multidecadal drought, approximately 1200- and 1000-years BP. Given regional variation in the timing and character of the collapse (Demarest et al. 2004; Hodell et al. 2007; Webster et al. 2007; Kennett and Beach 2014; Douglas et al. 2015), much remains to be discovered about the complex interactions between climate and society in the Maya lowlands." Lines 107-112.

In a second statement reviewer 2 commented:

"2. The statement that non-sedentary peoples aggregated to build public works with examples ("Stonehenge, Southern Arabia, Ohio, American Southwest") needs specific citations for each tradition and a direct link to Maya-area analogs if used to interpret Itzan."

To sharpen the discussion and more clearly articulate our conclusions regarding early Middle Preclassic trends at Itzan, I've rewritten the Event 3 paragraph and eliminated the reference to Stonehenge, Southern Arabia, Ohio, and the American Southwest.

Third, reviewer 2 observes,

"3. The text mentions a "pan-lowland Early Classic collapse"; this appears to conflate the widely discussed Terminal Preclassic (c. 150–250 CE) with Early Classic dynamics. Please verify terminology, provide appropriate citations, and correct if you mean Terminal Preclassic."

At Itzan, Terminal Preclassic deposits (as identified by ceramics) are difficult to discern—perhaps because of the limited nature of my excavations. What is obvious at Itzan, and at the nearby Late Preclassic center of Chak Akal, is the near absence of Early Classic materials—a trend also found coevally at Ceibal and Altar de Sacrificios. Rather than referring to a geographically broad Early Classic hiatus, as I did previously, I now situate Early Classic trends at Itzan within the broader context of those found at other major drainage settlements:

"This hiatus in coprostanol and PAH input coincides with a Terminal Preclassic through Early Classic decline at nearby Seibal (Willey et al 1975; Inomata et al. 2017) and Altar de Sacrificios (Smith 1972). At Itzan and the neighboring Chak Akal (Johnston 2006), this is evident archaeologically in an almost complete absence of Early Classic ceramics. "Lines 149-152.

Fourth, the reviewer's comments about the pan-lowland repercussions of Late and Terminal Classic societal upheaval in the central lowlands implies a need for appropriate and illustrative references:

"4. Societal upheaval in the central lowlands, including inter-polity warfare, the collapse of royal dynasties and regional political hierarchies, population dislocations, and the disruption of regional economies and inter-regional exchange, almost certainly would have had significant pan-lowland repercussions."

I supply these references as follows: "

"Societal upheaval in the central lowlands and in the Petexbatun polity, including inter-polity warfare, the collapse of royal dynasties and regional political hierarchies, population dislocations, and the disruption of regional economies and inter-regional exchange (Demarest 2004), almost certainly would have had significant pan-lowland repercussions." Lines 236-240.

**Similarly,**

"This period was a time of major societal restructuring across the lowlands, hypothesised to be in response to a regional drought, although there is no significant shift in the δDlw record (Hodell et al. 1995; Rosenmeier et al. 2002; Keenan et al. 2021)." Lines 171-173.

Finally, I greatly appreciate the fact that reviewer 2 took the time to generously prepare a full list of appropriate references. Thank you.

Thank you to the second reviewer for their review, including suggestions for improvement of the figures, which we accept. BCE/CE has been added to Figures 2 and 3 and some more detail has been added for both captions.

"In the 3.2 section, further elaboration might be needed on how the precipitation and climate variability inferred from the Itza data at small, local spatial scales can be meaningfully extended to the broader regional context of the southwest lowlands".

The climate signals recorded by plant waxes are expected to be broadly regional in scale, as opposed to land use/vegetation signals, because climate changes (i.e. droughts or increased in precipitation) are an atmospheric process that is not limited to individual catchments. In fact, it is surprising that the signal is so different from relatively nearby sites (e.g. Salpeten) given that we expect broadly similar patterns. Though we can't state with certainty that it applies across the southwest lowlands, it is suggestive that the pattern of climate change was different in this region. For example, a record from the highlands (Lake Kail; Stansell et al. 2020) shows no sign of Terminal Classic drought.

**Reviewer 3**

We thank reviewer 3 for their review. The request to cite a wider number of climate records from the lowlands has been satisfied, in addition to the references suggested by the other two reviewers.

Reviewer 3's point "Another criticism is that they do not consider the watershed linkages to this lake. Their records show pulses of five events or peaks that provide interesting nuances to human-climate-environmental interactions, but are these only due to the general land use and climate changes or due to sediment delivery and the magnitude and frequency questions of geomorphology?" has been addressed in Keenan et al. (2021), the paper applying fecal stanols to the Itzan sediment core. This paper shows how there was not a clear relationship between the biomarker data and the sedimentology (namely magnetic susceptibility). This implies that the variation was not sedimentological in nature. Further, the isotopic data is unlikely to be affected by sedimentology, as it is a ratio and not directly impacted by sediment delivery rates. A line has been added (258-260) referring to this.

Keenan, B. et al. Molecular evidence for human population change associated with climate events in the Maya lowlands. Quat. Sci. Rev. 258, (2021).